# Moving Object Detection and Tracking by Event Frame from Neuromorphic Vision Sensors

**DOI:** 10.3390/biomimetics7010031

**Published:** 2022-02-27

**Authors:** Jiang Zhao, Shilong Ji, Zhihao Cai, Yiwen Zeng, Yingxun Wang

**Affiliations:** School of Automation Science and Electrical Engineering, Beihang University, Beijing 100191, China; jzhao@buaa.edu.cn (J.Z.); shilongji@buaa.edu.cn (S.J.); zy1803102@buaa.edu.cn (Y.Z.); wangyx@buaa.edu.cn (Y.W.)

**Keywords:** neuromorphic vision sensors, event frame, object detection, object tracking

## Abstract

Fast movement of objects and illumination changes may lead to a negative effect on camera images for object detection and tracking. Event cameras are neuromorphic vision sensors that capture the vitality of a scene, mitigating data redundancy and latency. This paper proposes a new solution to moving object detection and tracking using an event frame from bio-inspired event cameras. First, an object detection method is designed using a combined event frame and a standard frame in which the detection is performed according to probability and color, respectively. Then, a detection-based object tracking method is proposed using an event frame and an improved kernel correlation filter to reduce missed detection. Further, a distance measurement method is developed using event frame-based tracking and similar triangle theory to enhance the estimation of distance between the object and camera. Experiment results demonstrate the effectiveness of the proposed methods for moving object detection and tracking.

## 1. Introduction

Intelligent agents such as robots, unmanned aerial vehicles (UAVs), unmanned ground vehicles (UGVs), and autonomous underwater vehicles (AUVs), are widely used in military and civilian fields [1,2,3,4,5,6,7,8]. Object detection and tracking are very important to improve the autonomy of intelligent agents. The complex environment and fast-moving object bring many challenges to object detection and tracking, and many researchers have focused on the detection and tracking of moving objects.

Object detection methods can be divided into two categories: traditional object detection methods [9,10,11,12,13,14,15,16] and deep learning-based object detection methods [17,18,19,20,21,22,23]. For traditional object detection methods, Viola and Jones [10,11] propose an algorithm that realizes the real-time detection of human faces for the first time. Their algorithm is called the VJ detector. The VJ detector uses a sliding window for detection and greatly improves the detection speed by using three technologies: the integral map, feature selection, and detection cascades [9,10,11]. Dalal et al. propose a histogram of oriented gradient (HOG) feature descriptor [12]. HOG is an important improvement to scale-invariant feature transformation [13,14] and shape contexts [15]. The HOG detector is also an important foundation for many object detectors. Felzenszwalb et al. [16] propose the deformable parts model (DPM) algorithm, which is a milestone for traditional object detection algorithms. DPM follows the detection principle of “divide and conquer” [9]. In this principle, the training can be simply regarded as learning the correct way to decompose an object, and the inference can be regarded as a collection of detecting different object parts [9,16]. Since AlexNet won the ImageNet competition championship, more and more studies have focused on object detection by deep learning. For deep learning-based methods, Girshick et al. [20] propose regions with convolutional neural network features (RCNN) for object detection. RCNN improves the detection accuracy a lot, but the redundant feature calculation makes the detection speed very slow [9,20]. He et al. [21] propose spatial pyramid pooling networks (SPPNet). SPPNet introduces the SPP layer, which makes it faster than RCNN by more than 20 times, and the accuracy is almost unchanged [21]. Ren et al. [22] propose the Faster RCNN detector. Faster RCNN introduces the regional proposal network (RPN), making it the first near real-time deep learning detector [9,22]. Liu et al. [23] propose a single shot multibox detector (SSD). SSD is a one-stage object detection algorithm. It introduces multi-reference and multi-resolution detection technology [9], which greatly improves the detection accuracy of the one-stage detection algorithm.

Object tracking can reduce the missed detection in object detection. The main object tracking algorithms include correlation filter tracking [24,25,26,27,28,29] and non-correlation filter tracking [30,31,32,33,34]. For correlation filter tracking algorithms, Bolme et al. [27] propose the minimum output sum of squared error (MOSSE) filter. MOSSE changes the situation that correlation filter tracking algorithms are not suitable for online tracking [24,27]. Ma et al. [28] propose rich hierarchical convolutional features in a correlation filter (CF2) for visual tracking. Qi et al. [29] propose the hedged deep tracking (HDT) algorithm, which combines multiple weak trackers to form a strong tracker. For non-correlation filter tracking algorithms, Zhang et al. [33] propose the convolutional networks without training (CNT) algorithm, which can utilize the inner geometry and local structural information of the object. CNT can adapt to changes in the object’s appearance during tracking [24,33]. The structural sparse tracking (SST) algorithm [34] uses the inherent relationship between the local object patches and the global object to learn sparse representation together. The object location is estimated based on the object dictionary template and the corresponding block with the largest similarity score from all particles [30,32].

There are some defects in current object detection and tracking methods. On the one hand, the fast movement of the object may lead to motion blur, which affects the results of object detection and tracking. On the other hand, the change in illumination may cause instability in object detection and tracking. The main contribution of this paper can be summarized as follows:
This paper proposes a detection method that combines the event frame from neuromorphic vision sensors and the standard frame to improve the effect of fast object movement or large changes in illumination.It uses the improved kernel correlation filter (KCF) algorithm for event frame tracking to solve the problem of missed detection.It proposes an event frame-based distance measurement method to obtain the distance information of the object.

The remainder of this paper is organized as follows. In Section 2.1, the preliminaries are introduced, and the problem formulation is stated. In Section 2.2, Section 2.3, Section 2.4, Section 2.5 and Section 2.6, the combined detection and tracking method is introduced in detail, including the event frame pre-processing, combined detection, event frame tracking, and object distance measurement. In Section 3, experiment results and discussions are presented. Section 4 summarizes the contribution of this paper and presents future work.

## 2. Materials and Methods

### 2.1. YOLO Algorithm

You only look once (YOLO) is a typical one-stage object detection algorithm [35,36,37,38]. Most traditional object detection algorithms have problems such as low sliding window efficiency and insufficient feature robustness [9]. The object detection method based on deep learning can solve these problems and has become the main method of object detection [17,18,19]. From the aspect of detection stages, deep learning-based object detection methods can be divided into two-stage methods and one-stage methods [17,18,19]. Compared with the two-stage methods, the one-stage methods do not have an obvious candidate box extraction process and treat the object detection task as a regression problem [39,40,41]. One-stage methods directly input an entire image into the neural network and then predict the coordinates of the detection frame and the category and confidence of the object [39]. Therefore, the one-stage detection process is more simplified, and the detection speed is faster [39,40,41].

The structure of the YOLO algorithm is shown in Figure 1. First, YOLO changes the size of the image. Second, YOLO puts the image into the convolutional neural network. Finally, YOLO suppresses the non-maximum value to obtain the detection result.

The YOLO algorithm divides the input image into an S×S grid. Each grid cell predicts B bounding boxes and confidence scores. The mathematical relation between Confidence(*C*), Pr(obj) (*P(o)*), and IOU(*I*) is shown in Equation (1).
(1)C=P(o)×I

If no object exists in that grid cell, the Pr(obj) is 0. Otherwise, the Pr(obj) is 1. Each bounding box has 5 predictions: *x*, *y*, *w*, *h*, and confidence. The *x* and *y* represent the coordinates of the center position of the bounding box of the object predicted by the current grid. The *w* and *h* are the width and height of the bounding box. Each grid cell predicts a conditional class probability $Pr(Class_{i}|obj)\;(P(c|o))$. The conditional class probability represents the probability that an object belongs to a certain class. YOLO multiplies the conditional class probabilities and the individual box confidence predictions, as shown in Equation (2).
(2)P(c|o)×P(o)×I=P(c)×I 

In this way, the confidence of the specific class of each bounding box can be obtained. The multiplication result not only reflects whether there are objects in the bounding box, but also contains the probability information of the predicted class in the bounding box and the accuracy of the bounding box prediction.

### 2.2. Framework of Event Frame-Based Object Detection and Tracking

Detection capability is usually required for intelligent agents, such as UAVs, to perform complex tasks. Figure 2 shows an example of object detection and tracking. It may have motion blur when the object is moving at a high speed, causing the object detection algorithm to fail. Compared with the standard cameras, the event cameras have less motion blur and are easier to detect high-speed moving objects. However, object detection based on event cameras is not effective when the object moves slowly. The combined detection using event frame and standard frame can leverage the benefits of both cameras, making object detection suitable for more scenes. Missed detection may occur during object detection, and therefore, we add event frame-based object tracking to reduce missed detection. Finally, a distance measuring algorithm is used to estimate the distance between the object and the camera.

The framework of event frame-based object detection and tracking scheme proposed in this paper is shown in Figure 3. First, we obtain event streams from the event camera. The event frame reconstruction algorithm accumulates event streams in a certain period of time, and then the event frame can be obtained. Second, we use the noise filtering algorithm to remove the noise in the event frame. Third, we run the combined detection algorithm to get the position of the object in the frame. Further, we use the improved KCF algorithm to track the position of the object. Finally, we run the distance measurement algorithm to calculate the distance between the object and the event camera.

### 2.3. Event Frame Pre-Processing

#### 2.3.1. Event Frame Reconstruction

Traditional object detection methods cannot process event streams directly. Therefore, we need to transform event streams into event frames that are similar to traditional standard frames.

The event camera continuously provides ON events and OFF events with timestamp marks. We can collect events within a certain period of time, $\Delta t=\left[t_{start}:t_{end}\right]$, and insert these events into a frame. After this period of time, the previous event frame is closed, and the generation of the next event frame starts. The definition of event frame is shown in Equation (3).
(3)$$E=\int_{t_{s}}^{t_{e}}E_{xy}\left(t_{c}\right)dt_{c}$$
where $E_{\;}$ represents event frame, $t_{s}$ is the start time, $t_{e}$ is the end time, $t_{c}$ is the current time, and $E_{xy}$ represents event triggered at coordinate (x,y).

The event frame reconstruction method used in this paper treats ON events and OFF events equally. If a pixel triggers an event in a time period, Δt, the pixel in the frame is drawn as a black pixel no matter whether it triggers an ON event or an OFF event and no matter how many events are triggered, and the background of the event frame remains white. Finally, the event frame is a black-and-white picture, in which the black part is where the event is triggered, as shown in Figure 4. The generated event frame has some noises, which need to be removed by the filter later.

#### 2.3.2. Noise Filtering for Dynamic Vision Sensors

This paper uses a filtering method known as the nearest neighbor filter. In the event frame, if the number of the events around an event is less than a given threshold, the event is considered to be noise, and it is removed. The events that can pass the filter are defined as Equation (4).
(4)$$F_{N}=\{\;E_{i}|\;N\left(E_{i}\right)\;\geq \;L\;\}{\;}_{\;}$$
where $F_{N}$ represents a collection of events that can pass the filter, $E_{i}$ represents the *i*-th event, and $N\left(E_{i}\right)$ is the number of events around the *i*-th event within 8 pixels. L is the given threshold, which is set to be 1.

Figure 5 shows the event frame obtained when the event camera shoots a still scene. Figure 5a is the event frame before filtering, and there are some noise points in the picture, which are circled in red. After filtering, the noise points are removed, as shown in Figure 5b.

Figure 6 shows the event frame obtained when the event camera shoots the upper body of a moving person. Figure 6a is the event frame before filtering, and there are some noise points on the right side of the picture. After filtering, the noise points are removed, as shown in Figure 6b.

From Figure 5 and Figure 6, we can see that the nearest neighbor filtering algorithm has a good denoising ability for pictures of both stationary objects and moving objects with the neuromorphic vision sensor.

### 2.4. Combined Detection Based on Event Frame and Standard Frame

#### 2.4.1. Detection Based on Probability

When the camera or object is moving at a high speed, the effect of object detection based on the event camera will be better than that by a standard camera. When the camera or object is moving very slowly, the effect of object detection based on a standard camera will be better than that by an event camera. In order to achieve a good detection effect in both high-speed movement and slow movement, we combine the detection results of the event camera with the detection results of the standard camera.

After the event frame and standard frame are detected by YOLOv3, the position of the object on the frame and the probability of the object belonging to a certain class can be obtained. By comparing the probability of the two detection results, we can obtain the result of combined detection. When the probability of the event frame after the detection is greater than the probability of the standard frame after the detection, the detection result of the event camera is used as the combined detection result. When the probability of the standard frame after the detection is greater than the probability of the event frame after the detection, the detection result of the standard camera is used as the combined detection result.

The framework of combined detection based on probability is shown in Figure 7. After the event frame is put into the convolutional neural network (CNN), the probability is obtained when the object belongs to a certain class. Here, we abbreviate the probability of event frame-based detection as $P_{e}$. At the same time, the standard frame is put into the convolutional neural network to get the probability when the object belongs to a certain class, and we abbreviate it as $P_{s}$. Then, the algorithm compares $P_{e}$ and $P_{s}$. If $P_{e}$ is greater than $P_{s}$, we use the result of the event frame detection as the result of combined detection. If $P_{s}$ is greater than $P_{e}$, we use the result of the standard frame detection as the result of combined detection.
(5)$$P_{e}=P{\left(c\right)}_{e}{}_{\;}$$
(6)$$P_{s}=P{\left(c\right)}_{s}{}_{\;}\;$$
where $P_{e}$ represents the probability when an object belongs to a certain class after event frame-based detection. $P_{s}$ is the probability when an object belongs to a certain class after standard frame-based detection.

#### 2.4.2. Detection Based on Color

When there is more than one person in the frame, it may be necessary to detect a specific person among them. The frame of the event camera contains less information, and it is hard to distinguish different people using the event frame only. On the basis of using the event camera for detection and using a standard camera to judge the color of the object, we can distinguish persons dressed in different colors without adding too much calculation.

The framework of combined detection based on color is shown in Figure 8. After preprocessing the event frame, we use YOLOv3 to detect the event frame to obtain the object class and the object position on the frame. At the same time, we obtain the RGB (red, green, blue) frame from the standard camera and convert the image from the RGB color space to the HSV (hue, saturation, value) color space. Then, the HSV frame is binarized. The black part of the frame will become white pixels, and the rest will become black pixels. Send the object position of the event frame after object detection to the binarized frame, and count the number of white pixels in the object area in the binarized frame. If the number of white pixels divided by the number of pixels in the entire object area exceeds a certain threshold, the object is considered to be detected. The threshold is set to be 0.3. Otherwise, we consider the object to be the wrong object. This article defines the ratio of the number of white pixels in the detection box to the total number of pixels as *T*, and the calculation formula is shown in Equation (7).
(7)$$T=\frac{p_{w}}{p_{a}}$$
where $p_{w}$ is the number of white pixels in the object detection box, and $p_{a}$ is the number of all pixels in the detection frame.

### 2.5. Event Frame-Based Tracking by Improved KCF

KCF is a discriminative model tracking algorithm. First, KCF samples the object according to the position of the object specified in the first frame. Then, a training sample set with the object at different positions is constructed through cyclic shift, and different labels are assigned according to the distance between the object and the center of the image block. We trained a linear regression model based on the training sample set and its labels. Since the training sample set is obtained by cyclically shifting the image block in the first frame, the entire sample set is a cyclic matrix, and the elements in each column of the cyclic matrix are cyclically moved down by the elements in the previous column.

#### 2.5.1. Training Phase


(1)Linear regression


The KCF algorithm uses continuous labels to mark samples, and it assigns values between 0 and 1 according to the distance between the tracked object and the center of the selection box. The closer the object is to the center, the closer the label is to 1. The farther away the object is from the center, the closer the label is to 0. The mathematical relation can be characterized by a Gaussian function or a sine function.

Assuming that given some training samples and their expected output values, the ultimate goal of training is to find a function $f\left(z\right)=w^{T}z$ that minimizes the cost function:(8)$$\min\underset{i}{\sum }{\left(f\left(x_{i}\right)-y_{i}\right)}^{2}+\lambda {∥w∥}^{2}$$

Write the above formula in matrix form:(9)$$\min{∥Xw-y∥}^{2}+\lambda {∥w∥}^{2}$$

Let the derivative be 0, and we obtain the following formula:(10)$$w={\left(X^{T}X+\lambda I\right)}^{-1}X^{T}y$$


(2)Linear regression under discrete Fourier transform


The sample set in the KCF algorithm is obtained by shifting the image blocks collected in the initial frame. The entire training set is constructed from one sample
(11)$$PX={\left[x_{1},x_{2},\ldots x_{n-1}\right]}^{T}$$
where $X={\left[x_{1},x_{2},\ldots x_{n}\right]}^{T}$ and $P=\left[\begin{array}{ccccc} 0 & 0 & \cdots & 0 & 1\\ 0 & 1 & \cdots & 0 & 0\\ 0 & 0 & \cdots & 0 & 0\\ \vdots & \vdots & \vdots & \vdots & \vdots \\ 0 & 0 & \cdots & 1 & 0 \end{array}\right]$.

Therefore, the entire training set is a cyclic matrix, and each row vector of the cyclic matrix is obtained by shifting the previous row to the right. The circulant matrix has a property that can be diagonalized by the Fourier matrix:(12)$$X=Fdiag\left(\hat{x}\right)F^{H}$$
(13)$$\hat{x}=F\left(x\right)=\sqrt{n}FX$$
(14)$$w=C\left(F^{-1}\left(\frac{{\hat{x}}^{*}}{{\hat{x}}^{*}⊙\hat{x}+\lambda }\right)\right)y$$

Using the circulant matrix convolution property, we obtain
(15)$$\hat{w}=\frac{\hat{x}⊙\hat{y}}{{\hat{x}}^{*}⊙\hat{x}+\lambda }$$


(3)Linear regression in kernel space


The basic idea of the kernel is to make the transformed data linearly separable through a nonlinear mapping function ϕ(x). Then, the linear model $f(x_{i})=w^{T}\phi \left(x\right)$ can be used to fit the functional relationship in the transformed space. Therefore, the weight term obtained is as follows:(16)$$w=\underset{w}{\min}{∥\phi \left(X\right)-y∥}^{2}+\lambda {∥w∥}^{2}$$

W can be represented linearly by the row vector of $\phi \left(X\right)={\left[\phi \left(x_{1}\right),\phi \left(x_{2}\right),\ldots ,\phi \left(x_{n}\right)\right]}^{T}$, and we can make $w=\underset{i}{\sum }\alpha_{i}\phi \left(x_{i}\right)$. The above formula becomes:(17)$$\alpha =\underset{\alpha }{\min}{∥\phi \left(X\right)\phi {\left(X\right)}^{T}\alpha -y∥}^{2}+\lambda {∥\phi {\left(X\right)}^{T}\alpha ∥}^{2}$$

Let the derivative of α be 0.
(18)$$\alpha ={\left(\phi \left(X\right)\phi {\left(X\right)}^{T}+\lambda I\right)}^{-1}y$$

Let *K* denote the kernel matrix, which can be calculated by the kernel function as $K=\phi \left(X\right)\phi {\left(X\right)}^{T}$, and then $\alpha ={\left(K+\lambda I\right)}^{-1}y$. We diagonalize the circulant matrix *K* to get:(19)$$\alpha =Fdiag{\left({\hat{K}}^{xx}+\lambda \right)}^{-1}F^{H}y$$
(20)$$\hat{\alpha }=\frac{\hat{y}}{{\hat{K}}^{xx}+\lambda }$$

#### 2.5.2. Detecting Phase


(1)Fast detection


Under normal circumstances, a regression calculation is inefficient for a sample. Usually, the candidate image block samples are tested to select the closest one to the initial sample. The acquisition of these candidate image blocks is constructed by shifting a sample for fast detection.

$K^{Z}=\phi \left(X\right)\phi {\left(Z\right)}^{T}$ represents the kernel matrix between the sample set used for training and all candidate image patch sets. Because the sample set and the candidate image set are constructed by the basic sample *x* and the basic image block *z*, respectively, each element of $K^{Z}$ is given by $\kappa \left(P^{i-1}z,P^{j-1}x\right)$, and is cyclic for the appropriate kernel function.
(21)$$f\left(z\right)={\left(K^{z}\right)}^{T}\alpha =F^{H}diag\left({\hat{k}}^{XZ}\right)F\alpha$$
(22)$$\hat{f}\left(z\right)={\hat{k}}^{XZ}⊙\hat{\alpha }$$


(2)Fast calculation of kernel matrix


Although there are faster training and detection algorithms, they still rely on computing a core correlation. Kernel correlation includes the kernel that calculates all relative displacements of two input vectors. This represents another computing bottleneck because the naive evaluation of n kernels of n signals will have quadratic complexity. However, using the cyclic shift model will allow us to effectively utilize redundancy in this costly calculation.

The polynomial kernel matrix can be expressed as $k_{i}^{xx^{\prime }}$.
(23)$$k_{i}^{xx^{\prime }}=g{\left(F^{-1}\left({\hat{x}}^{*}⊙{\hat{x}}^{\prime }\right)\right)}^{T}$$

Therefore, for the polynomial kernel *x*, we have:(24)$$k^{xx^{\prime }}={\left({\left(F^{-1}\left({\hat{x}}^{*}⊙{\hat{x}}^{\prime }\right)+a\right)}^{b}\right)}^{T}$$

In addition, radial basis kernel functions, such as Gaussian kernels, are functions of ${∥x_{i}-x_{j}∥}^{2}$. Given that ${∥x_{i}-x_{j}∥}^{2}={∥x_{i}∥}^{2}+{∥x_{j}∥}^{2}-2x_{i}{}^{T}x_{j}$, Equation (25) can be obtained.
(25)$$k^{xx^{\prime }}=g{\left({∥x∥}^{2}+{∥x^{\prime }∥}^{2}-F^{-1}\left({\hat{x}}^{*}⊙{\hat{x}}^{\prime }\right)\right)}^{T}$$

For the Gaussian kernel, there is Equation (26).
(26)$$k^{xx^{\prime }}=\exp(-\frac{1}{\sigma^{2}}\left({∥x∥}^{2}+{∥x^{\prime }∥}^{2}-F^{-1}\left({\hat{x}}^{*}⊙{\hat{x}}^{\prime }\right){)}^{T}\right)$$

Since the original KCF cannot judge object loss and the tracking effect is not good when occluded, the KCF algorithm is modified. Improved KCF determines whether the object is lost according to the peak response value. When the object is occluded, the algorithm inputs $x,y,v_{x},v_{y}$ at the current moment to the Kalman filter to estimate the object’s position at the next moment.

The state equation of the system is shown in Equation (27).
(27)$$x_{k}=A_{k}x_{k-1}+B_{k}u_{k}+w_{k}$$

The observation equation of the system is shown in Equation (28):(28)$$z_{k}=H_{k}x_{k}+v_{k}$$
where $x_{k}$ is the state value of the system at time *k*, $A_{k}$ is the system state transition matrix, $x_{k-1}$ is the state value of the system at time *k* − 1, $B_{k}$ is the control matrix, $u_{k}$ is the control quantity at time *k*, $w_{k}$ is the systematic error, $z_{k}$ is the measured value of the system at time *k*, $H_{k}$ is the measurement matrix, and $v_{k}$ is the measurement error.

The iterative process of the Kalman filter is mainly divided into two stages: the prediction stage and the update stage. In the prediction stage, the predicted state value $x_{k|k-1}$ and the minimum mean square error $P_{k|k-1}$ are required. The calculation formulas for both are shown in Equations (29) and (30).
(29)$$x_{k|k-1}=A_{k}x_{k-1|k-1}+B_{k}u_{k}$$
(30)$$P_{k|k-1}=A_{k}P_{k-1|k-1}A_{k}{}^{T}+Q_{k}$$

In the update phase, the new state value $x_{k|k}$, Kalman gain $K_{k}$, and the updated minimum mean square error $P_{k|k}$ need to be calculated. The calculation formulas for the three are shown in Equations (31)–(33).
(31)$$x_{k|k}=x_{k|k-1}+K_{k}\left(z_{k}-H_{k}x_{k|k-1}\right)$$
(32)$$K_{k}=P_{k|k-1}H_{k}{}^{T}{\left(H_{k}P_{k|k-1}H_{k}{}^{T}+R_{k}\right)}^{-1}$$
(33)$$P_{k|k}=\left(I-K_{k}H_{k}\right)P_{k|k-1}$$

When the position of the detection box is obtained, it is put into the Kalman filter to estimate the position of the object at the next moment. When the object is not occluded, the tracking result of the KCF algorithm is used as the object position. When the object is occluded, the result of Kalman filter estimation is used as the position of the object. The framework of this method is shown in Figure 9.

### 2.6. Event Frame-Based Distance Measurement

The detection and tracking algorithm of the event camera can only obtain the position of the object on the image. It cannot obtain the distance between the object and the event camera. Based on similar triangle theory, this paper proposes an event-frame-based distance measurement algorithm to measure the distance between the event camera and object. As shown in Figure 10, the object is a person, and *O* is the optical center of the camera. The person on the right in Figure 10 is the object in the real world, and *w* is the height of the object in the real world. The person on the left in Figure 10 is the object on the imaging plane, and *p* is the height of the object on the image, which is equal to the height of the tracking box. The distance *f* between the imaging plane and the optical center is the focal length. The distance *d* between the object and the optical center is the physical quantity we need to calculate.

The distance is a function of the actual size of the object, the size of the pixels of the object in the image, and the focal length of the event camera. The calculation formula is shown in Equation (34).
(34)$$d=\frac{w\times f}{p}$$
where *d* is the distance between the object and the camera, *w* represents the actual size of the object, *f* is the focal length of the event camera, and *p* represents the pixel size of the object on the image.

This article uses the height of the person instead of the width because the width may change frequently during the movement of a person, but the height can almost be considered unchanged.

## 3. Results

In this section, we perform three sets of experiments to test the scheme for object detection and tracking, as well as distance measuring, as shown in Table 1. The first set of experiments is object detection experiments. The second set of experiments is tracking experiments. The tracking experiments compare the KCF algorithm based on event frame with the improved KCF algorithm based on event frame. The third set of experiments compares the effects of the PnP (Perspective-*n*-Point) algorithm based on event frame with the similar triangle algorithm based on an event frame, and tests the ranging effect of the similar triangle algorithm at different distances.

### 3.1. Object Detection

The object detection experiments consist of two parts, including experiments of combined detection based on probability and experiments of combined detection based on color.

#### 3.1.1. Combined Detection Based on Probability

In the experiments, a stationary person model is placed on the ground. We control the event camera and the standard camera to shoot the model from different angles and create a dataset. It contains 271 pictures, in which 244 pictures are used for training and 27 pictures are used for testing. After 100 rounds of training, the final mAP (mean average precision) reaches 0.993. Then, we control the event camera and the standard camera to shoot the model from different angles at different speeds and make another dataset that contains 121 pictures for the detection experiment.

Specifically, for event frame generation, we consider the polarity of the event and ignore trigger times. At first, the event frame is set to gray. In a certain time period, if the last event triggered by a pixel is an ON event, the color of the pixel is set to white. If the last event triggered by a pixel is an OFF event, the color of the pixel is set to dark gray. Finally, the event frame consists of a gray part, dark gray part, and white part, which indicates no event triggered, OFF event triggered, and ON event triggered, respectively. Other experimental parameters and dataset properties are shown in Table 2.

Figure 11 shows the results of combined detection when the cameras move very slowly. As shown in Figure 11a, when the camera moves very slowly, the event frame has very little information. At this time, the object cannot be detected with the event frame. The detection result of the standard frame is shown in Figure 11b, the standard camera can detect the object. The result of combined detection is shown in Figure 11c, and the probability $P_{e}$ obtained by event frame-based detection is less than the probability $P_{s}$ obtained by standard frame-based detection. Therefore, the detection result of standard frame is taken as the result of combined detection.

Figure 12 shows the results of combined detection when the cameras are moving fast. Figure 12a shows the result of the event frame-based detection. As we can see from the figure, the object can be detected. Figure 12b shows the detection result of the standard frame. Due to the fast movement, the standard frame has motion blur, and the object cannot be detected. The result of combined detection is shown in Figure 12c. Since the probability $P_{e}$ obtained by event frame-based detection is greater than the probability $P_{s}$ obtained by standard frame-based detection, the detection result of the event frame is taken as the result of combined detection.

Figure 13 shows the results of combined detection when the camera movement speed is not fast. Figure 13a is the result of the event frame-based detection. As we can see from the figure, the object can be detected. The detection result of the standard frame is shown in Figure 13b, and the standard camera can detect the object. The result of the combined detection is shown in Figure 13c. Since the probability $P_{e}$ obtained by event frame-based detection is greater than the probability $P_{s}$ obtained by standard frame-based detection, the detection result of the event frame is taken as the result of combined detection.

Table 3 shows the results of the three detection methods in a certain period of time. In all 121 frames of images, the event frame-based method detects 56 frames, the standard frame-based method detects 39 frames, and the combined method detects 87 frames. It can be seen that the success rate of combined detection is 55% higher than that of event frame-based detection and 123% higher than that of standard frame-based detection.

#### 3.1.2. Combined Detection Based on Color

The experimental scene is shown in Figure 14. There are two persons in different colors on the ground. The algorithm needs to detect the one that wears black. The other that wears green needs to be ignored.

The results are shown in Figure 15. Figure 15a is the detection result of the event frame-based detection algorithm. It cannot distinguish people wearing clothes of different colors, and the wrong object is detected. Figure 15b is the result of the combined detection algorithm. It can identify the person wearing black clothes and pants.

### 3.2. Object Tracking

The tracking experiments mainly compare the KCF algorithm and the improved KCF algorithm based on the event frame. In the experiments, the UAV with cameras keeps hovering, and a person runs from left to right. He meets trees on the way, which will block him. After that, he continues to run right, away from the trees, and is not blocked anymore. Overall, 215 pictures are collected, and the person is partly blocked or totally blocked in 20 of 215. Since the KCF algorithm does not need offline training, no extra dataset is needed. Other experimental parameters and dataset properties are shown in Table 4.

The results of the object tracking experiments are shown in Figure 16. The red box in the frame is the tracking result of the KCF algorithm, and the blue box is the tracking result of the improved KCF algorithm. Figure 16a shows the result when the person is not occluded. Both the KCF algorithm and the improved KCF algorithm can track the object. Figure 16b shows the tracking result when the person is occluded. The KCF algorithm fails to track, and the improved KCF algorithm can continue to track the person. Figure 16c also shows the tracking result when the person is occluded. The KCF algorithm fails to track, and the improved KCF algorithm can continue to track the person. Figure 16d shows the result when the person is not occluded. Both the KCF algorithm and the improved KCF algorithm can track the object.

Table 5 shows the results of the two object tracking methods in a certain period of time. In all 20 frames of images, the KCF algorithm tracks 13 frames, and the improved KCF algorithm tracks 20 frames. The success rate by the improved KCF algorithm is higher than that by the KCF algorithm when the object is blocked.

### 3.3. Distance Measurement

The distance measurement experiments consist of two parts. The first part is the comparison between the PnP distance measurement algorithm and the similar triangle distance measurement algorithm. In the second part, a similar triangle distance measurement algorithm is tested from different distances.

#### 3.3.1. Comparison of PnP and Similar Triangle Distance Measurements

The comparison of PnP distance measurement and similar triangle distance measurement results are shown in Figure 17. In the experiment, the hovering height of the UAV with cameras is 2.1 m, and the distance between the UAV and the person is about 10 m. Figure 17a is the result of the PnP distance measurement. The minimum value measured by PnP is 0.4 m, the maximum value measured is 34.8 m, and the average value is 17.6 m, which is quite different from the ground truth. Figure 17b is the result of a similar triangle distance measurement. The minimum value measured by the similar triangle is 9.4 m, the maximum value measured is 10.5 m, and the average value is 9.9 m, which is relatively close to the ground truth. Table 6 shows the results of the two distance measurement methods.

#### 3.3.2. Performance of Similar Triangle Distance Measurement

Figure 18 shows the results of similar triangle distance measurements at different distances. In the experiment, the hovering height of the UAV with cameras is 2.1 m, and the distance between the UAV and the person is about 10 m, 20 m, and 30 m. Figure 18a is the result of similar triangle distance measurement at a distance of 10 m. The minimum measured value is 9.4 m, the maximum measured value is 10.5 m, and the average value is 9.9 m, which is very close to the true value of 10 m. Figure 18b is the result of a similar triangular distance measurement at a distance of 20 m. The measured minimum value is 19.3 m, the measured maximum value is 21.5 m, and the average value is 20.4 m, which is very close to the true value of 20 m. Figure 18c is the result of a similar triangle distance measurement at a distance of 30 m. The minimum measured value is 27.8 m, the maximum measured value is 32.6 m, and the average value is 30.2 m, which is very close to the true value of 30 m. Table 7 shows the results of the similar triangle distance measurement method at different distances.

## 4. Conclusions

Based on the event frame from neuromorphic vision sensors and the standard frame, a new solution to detect and track moving objects is proposed in this paper. The experiment results show that the capacity of combined object detection method is stronger than that by using event frame or standard frame alone, and the combined method can also distinguish objects with different colors. We also propose an object tracking and distance measurement method based on an event frame. The experiment results of object tracking show that the event frame-based algorithm can track the object and deal with the problem of occlusion effectively. The experimental results of distance measurement show that the proposed method can obtain a more accurate distance between the object and camera. For future work, the following aspects will be considered: (1) to improve the filter algorithm for event frames to obtain high-quality event images; (2) to obtain experiment results in a more complex environment such as illumination changes.

## Figures and Tables

**Figure 1 biomimetics-07-00031-f001:**
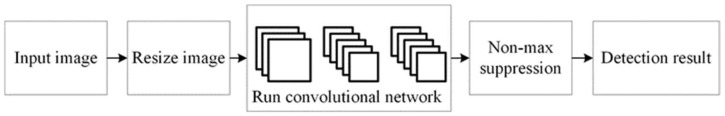
The structure of the YOLO detection algorithm.

**Figure 2 biomimetics-07-00031-f002:**
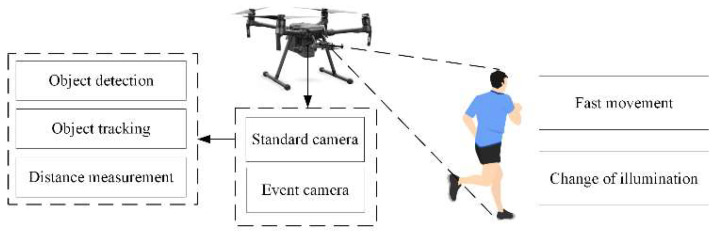
Example of object detection and tracking.

**Figure 3 biomimetics-07-00031-f003:**
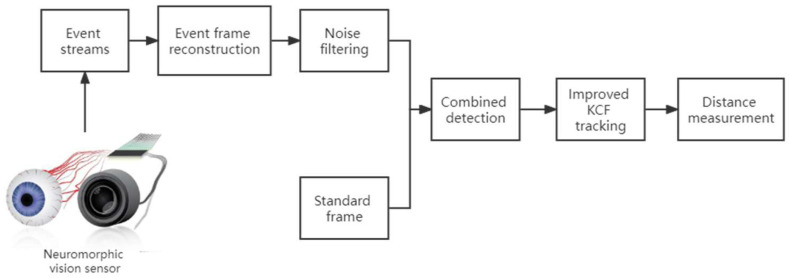
Framework of the event frame-based object detection and tracking scheme.

**Figure 4 biomimetics-07-00031-f004:**
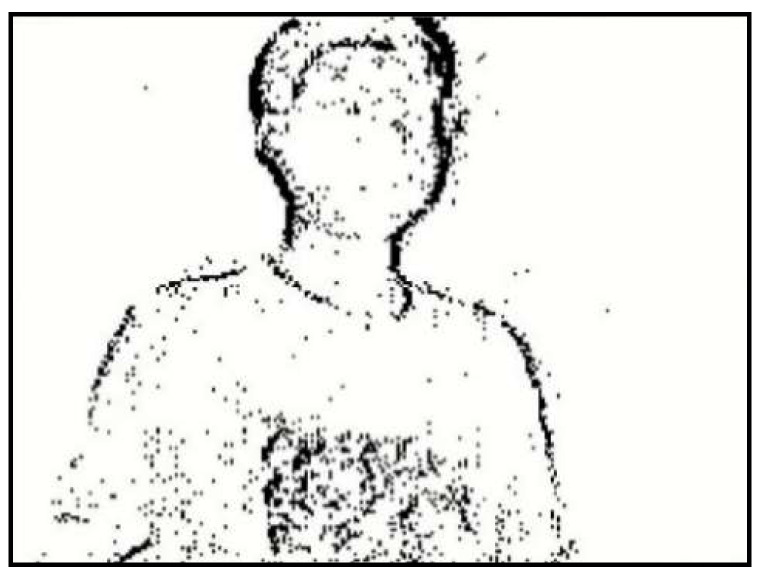
Event frame reconstruction.

**Figure 5 biomimetics-07-00031-f005:**
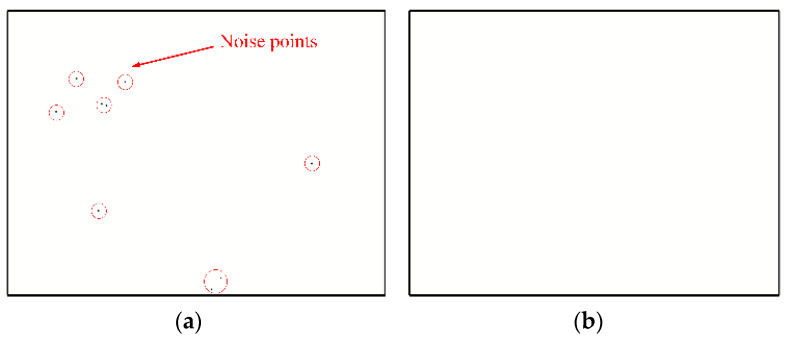
Results of nearest neighbor filter. (**a**) Before filtering, (**b**) After filtering.

**Figure 6 biomimetics-07-00031-f006:**
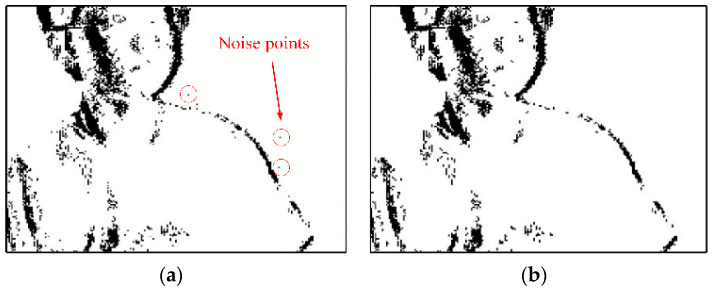
Results of nearest neighbor filter in a dynamic environment. (**a**) Before filtering, (**b**) After filtering.

**Figure 7 biomimetics-07-00031-f007:**
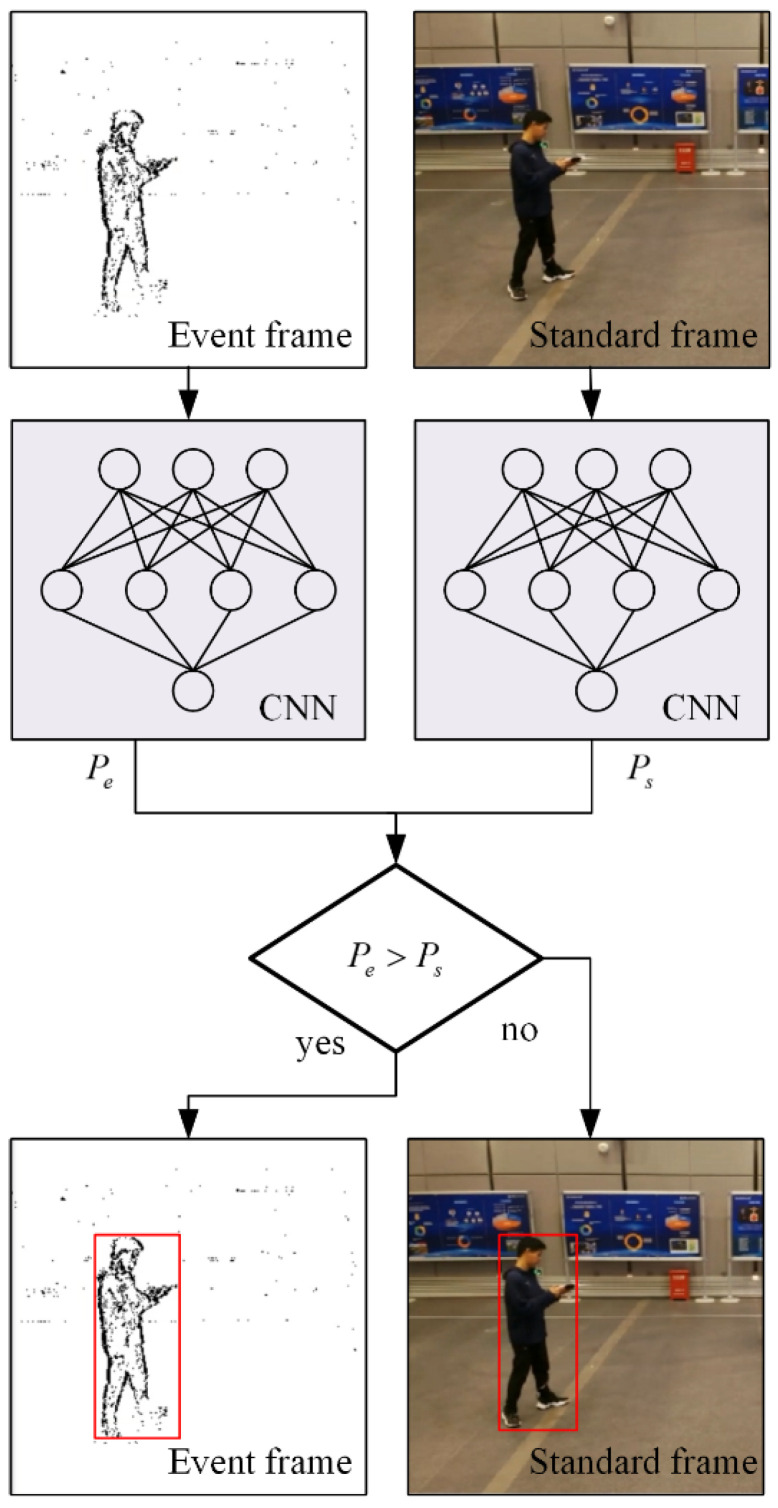
Framework of combined detection based on probability.

**Figure 8 biomimetics-07-00031-f008:**
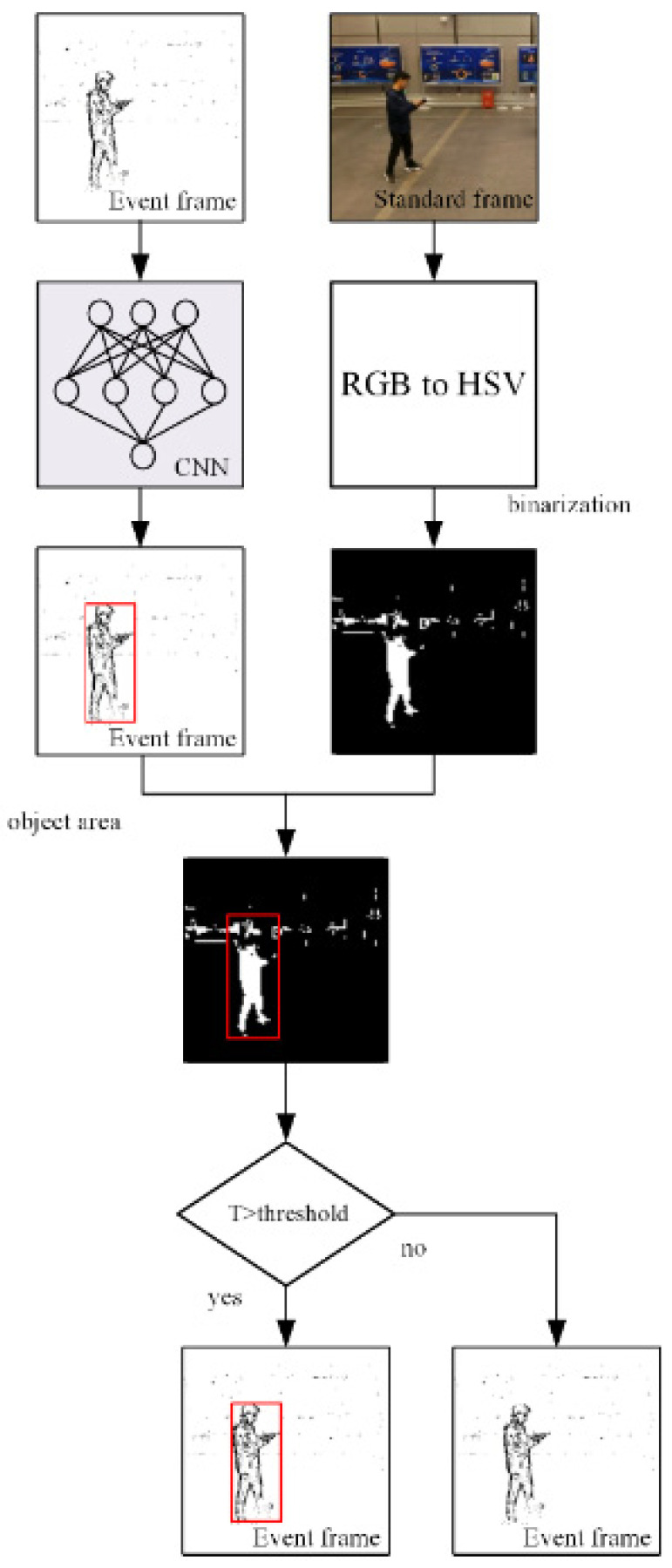
Framework of combined detection based on color.

**Figure 9 biomimetics-07-00031-f009:**
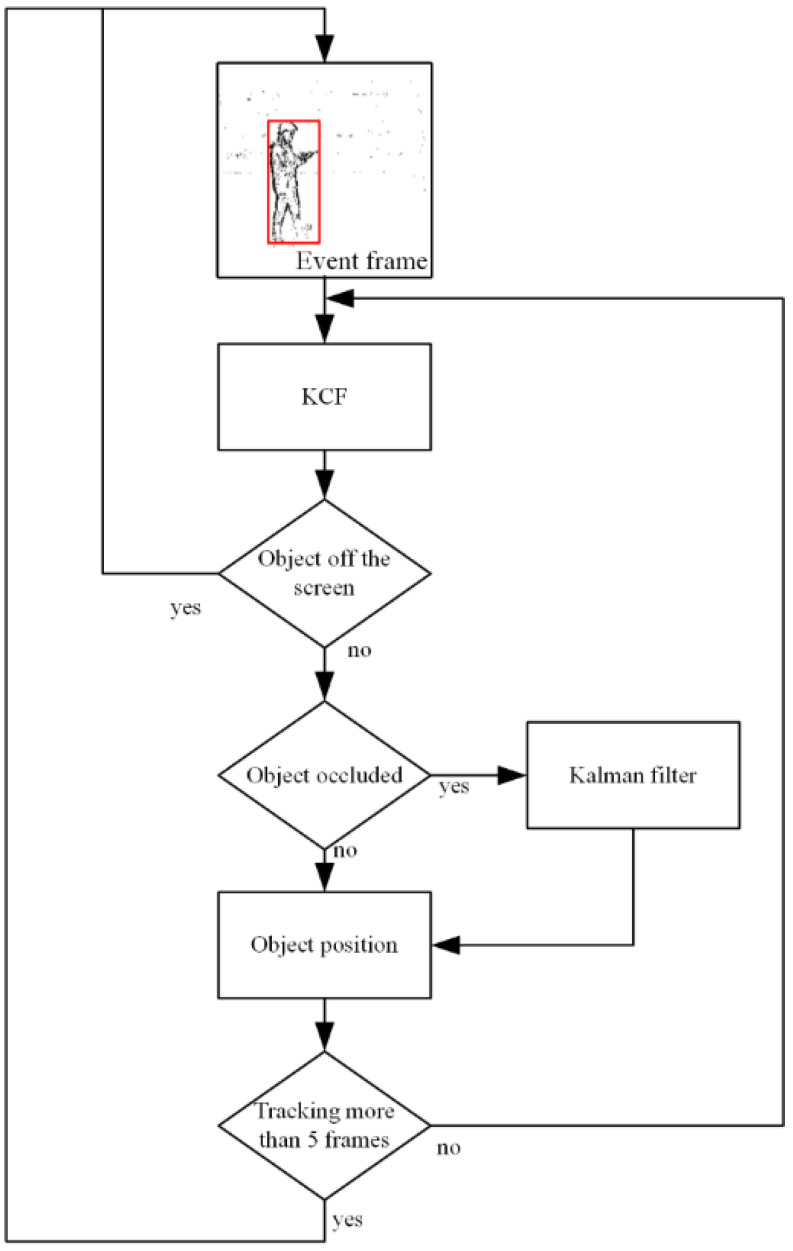
Flow chart of event-based tracking by improved KCF.

**Figure 10 biomimetics-07-00031-f010:**
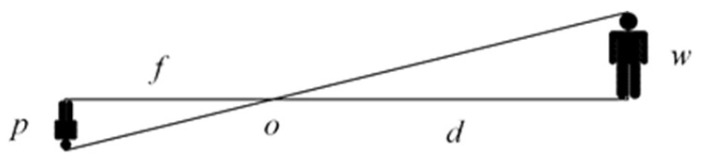
Principle of similar triangle distance measurement.

**Figure 11 biomimetics-07-00031-f011:**
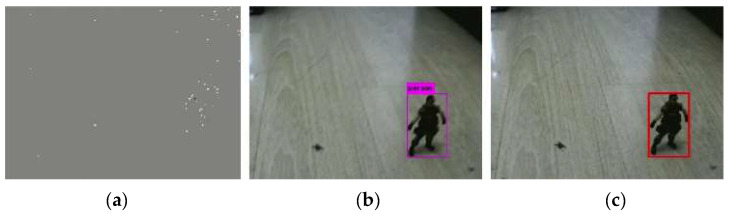
Results of combined detection (event frame: missed, standard frame: detected). (**a**) Event frame-based detection, (**b**) Standard frame-based detection, (**c**) Combined detection.

**Figure 12 biomimetics-07-00031-f012:**
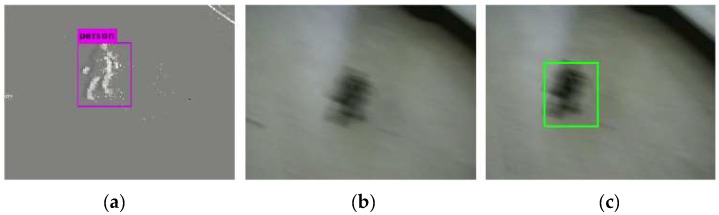
Results of combined detection (event frame: detected, standard frame: missed). (**a**) Event frame-based detection, (**b**) Standard frame-based detection, (**c**) Combined detection.

**Figure 13 biomimetics-07-00031-f013:**
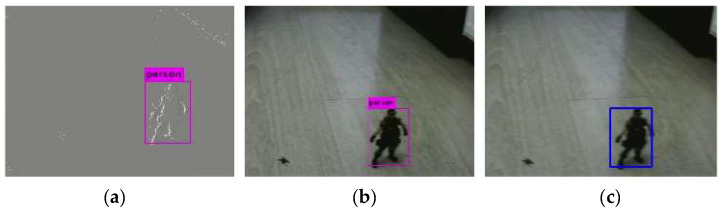
Results of combined detection (event frame: detected, standard frame: detected). (**a**) Event frame-based detection, (**b**) Standard frame-based detection, (**c**) Combined detection.

**Figure 14 biomimetics-07-00031-f014:**
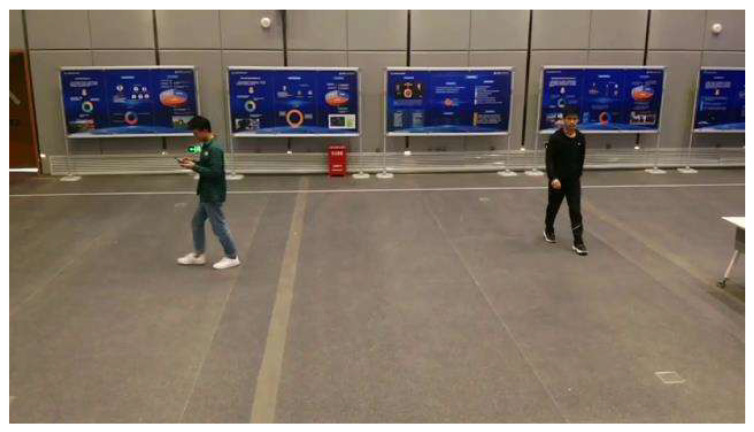
Experiment environment of combined detection based on color (left person: green clothes, right person: black clothes).

**Figure 15 biomimetics-07-00031-f015:**
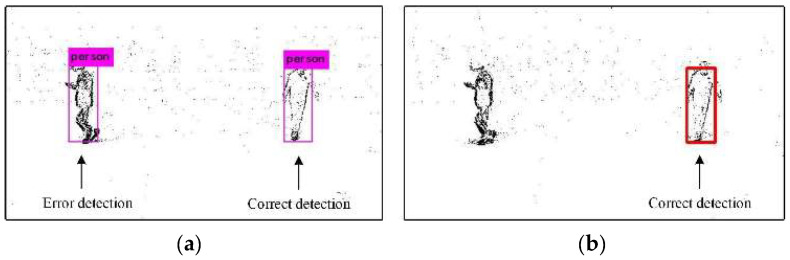
Comparison of event frame-based detection and combined detection based on color. (**a**) Event frame detection, (**b**) Combined detection based on color.

**Figure 16 biomimetics-07-00031-f016:**
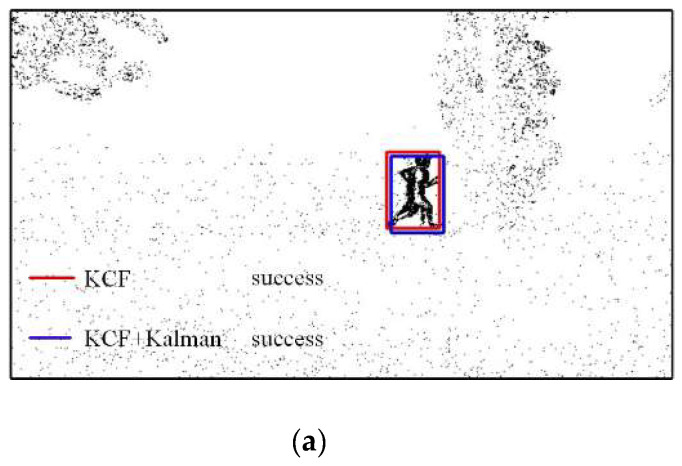
Comparison of object tracking results by KCF and improved KCF. (**a**) Frame 197, (**b**) Frame 205, (**c**) Frame 209, and (**d**) Frame215.

**Figure 17 biomimetics-07-00031-f017:**
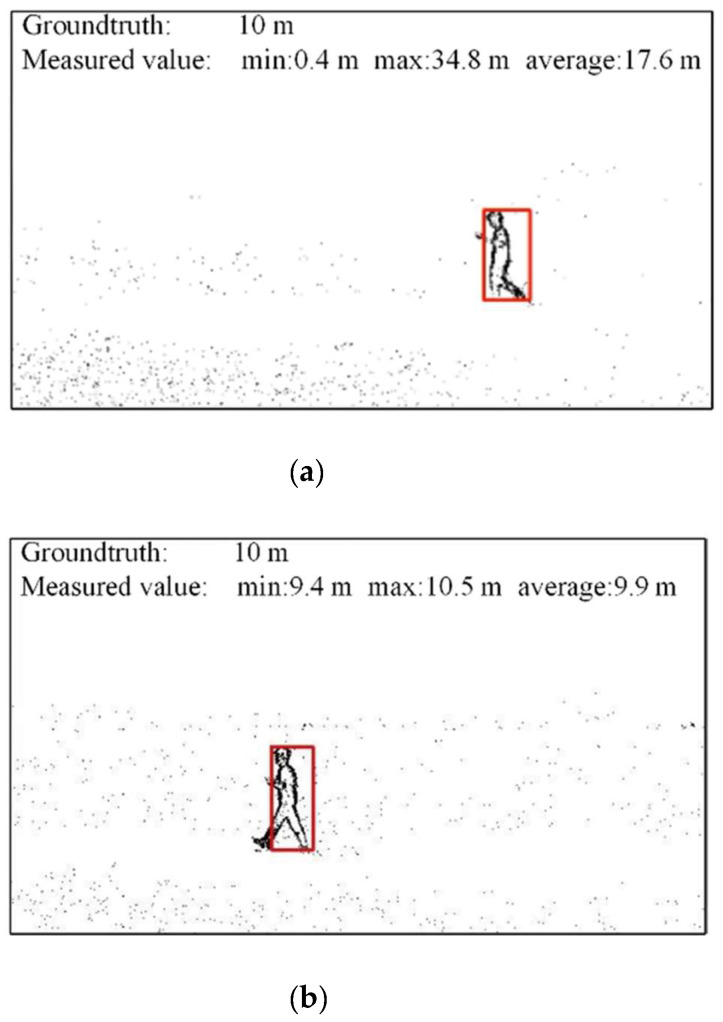
Comparison of distance measurement results by PnP and similar triangle. (**a**) PnP distance measurement, (**b**) similar triangle distance measurement.

**Figure 18 biomimetics-07-00031-f018:**
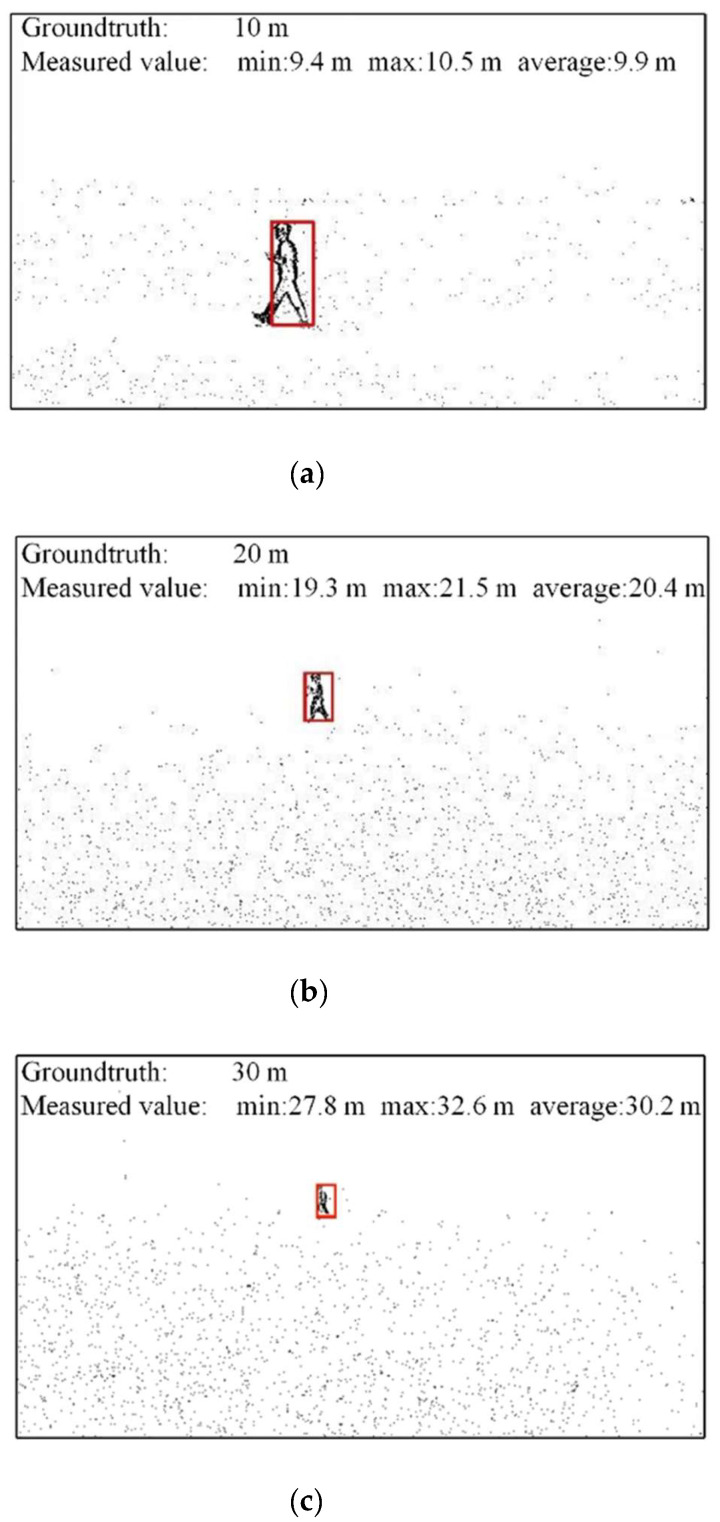
Results of a similar triangle distance measurement at different distances. (**a**) Groundtruth: 10 m, (**b**) Groundtruth: 20 m, (**c**) Groundtruth: 30 m.

**Table 1 biomimetics-07-00031-t001:** Three sets of experiments.

Nos.	Experiments Content
1	Object detection experiments
2	Tracking experiments
3	Distance measurement experiments

**Table 2 biomimetics-07-00031-t002:** Experimental parameters and dataset properties.

Parameters or Methods	Values or Implementations
Size of YOLOv3 dataset	271
mAP of YOLOv3	0.993
Iteration of YOLOv3	100
Size of detection experiment dataset	121
Event frame generating method	Consider event polarity, and ignore trigger times
Frame filter	nearest neighbor filter
Filter threshold *L*	1
Combine detection threshold *T*	0.3

**Table 3 biomimetics-07-00031-t003:** Numbers of frames detected by different methods.

Total Number of Frames	Number of Event Frame-Based Detection	Number of Standard Frame-Based Detection	Number of Combined Detection
121	56	39	87

**Table 4 biomimetics-07-00031-t004:** Experimental parameters and dataset properties.

Parameters or Methods	Values or Implementations
Size of tracking experiment dataset	215
Event frame generating method	Consider event polarity, and ignore trigger times
Frame filter	nearest neighbor filter
Filter threshold *L*	1
KCF peak value threshold	0.3

**Table 5 biomimetics-07-00031-t005:** Number of frames tracked by different object tracking methods.

Total Number of Frames	Number of Frames by KCF Algorithm	Number of Frames by Improved KCF Algorithm
20	13	20

**Table 6 biomimetics-07-00031-t006:** Results of PnP and similar triangle distance measurement methods.

Methods	True Values	Minimum Measured Values	Maximum Measured Values	Average Measured Values
PnP	10 m	0.4 m	34.8 m	17.6 m
Similar triangle	10 m	9.4 m	10.5 m	9.9 m

**Table 7 biomimetics-07-00031-t007:** Results of a similar triangle distance measurement method at different distances.

True Values	Minimum Measured Values	Maximum Measured Values	Average Measured Values
10 m	9.4 m	10.5 m	9.9 m
20 m	19.3 m	21.5 m	20.4 m
30 m	27.8 m	32.6 m	30.2 m

## Data Availability

Not applicable.

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
