# Peer review of "Moving Object Detection and Tracking by Event Frame from Neuromorphic Vision Sensors"

_biomimetics, 2022, doi:10.3390/biomimetics7010031_

Round 1

Reviewer 1 Report

This paper proposes joint utilization of standard & event cameras for object detection and tracking. Major revisions should be applied before resubmission.

1- The novelty of the paper is weak. The authors have used a combination of well-known methods without considering the computational complexity. Also, the proposed framework is not well unified.

2- The manuscript should be thoroughly re-organized. The paper does not have consistency almost in all sections.

3- All the details should be provided in the revised manuscript—for instance, all the details regarding the implementation, datasets, empirical experiments, etc. 

4- A native should proofread the paper. 

5- The claims should be experimentally proved, or the authors should cite some references. For instance, prove your claim in line 54.

Overall, the paper requires significant revisions in the context, style, grammar, figures, etc.

Author Response

Thank you very much for the valuable opinion of the manuscript. We appreciate your constructive remarks. The following parts list the detailed responses to your review comments. Please see the attachment.

Reviewer 2 Report

This paper proposes a new solution to moving object detection and tracking by using event frame from bio-inspired sensors. Qualitative results looks good, but I could not find the quantitative results compared to other state-of-the-art approaches. How good is your algorithm?

Also, the equation should be updated. It would be better to use C, o, A instead of Confidence, obj, and AE, The notations such as Confidence, obj, and AE are confusing. For example, AE looks like representing A times E.

Author Response

Thank you very much for the positive opinion on the manuscript. We appreciate your constructive remarks. The following parts list the detailed responses to your review comments. Please see the attachment.

Round 2

Reviewer 1 Report

The authors have majorly revised the paper. Although it is now much better, the English language and style are still poor. I believe proofreading the manuscript by a native person is necessary before publication.

Author Response

Thank you very much for the valuable opinion on the manuscript. We appreciate your constructive remarks. Please see the attachment.
